# Germline Variant Call Accuracy in Whole Genome Sequence Data from Canine Formalin-Fixed Paraffin-Embedded Tissue Samples

**DOI:** 10.3390/genes16111371

**Published:** 2025-11-11

**Authors:** Vidhya Jagannathan, Tosso Leeb

**Affiliations:** Institute of Genetics, Vetsuisse Faculty, University of Bern, 3001 Bern, Switzerland

**Keywords:** GATK, DeepVariant, WGS, dog, *Canis lupus familiaris*, animal, veterinary, methodology

## Abstract

Background/Objectives: Fresh frozen (FF) samples are routinely used to isolate high-molecular-weight intact genomic DNA. However, when FF samples are not available, archived formalin-fixed paraffin-embedded (FFPE) tissue samples often represent the only available material in clinical research. Due to formaldehyde-induced degradation of nucleic acids they pose special challenges for genetic investigations. In this study we compare whole-genome sequencing results on intact DNA versus fragmented DNA derived from FFPE samples of three dogs. Methods: We prepared matched libraries from FF and FFPE samples of three dogs affected by an inherited disease, *EFNB3*-related congenital mirror movement disorder 1 (CMM1). Paired-end short-read sequencing data were obtained on an Illumina sequencer and analyzed with adapted workflows for FF or FFPE data, respectively. Results: The data between FF and FFPE samples were largely consistent. FF data showed a superior variant call accuracy, as expected. However, the data quality from the FFPE samples was sufficient to correctly identify the causal variant in *EFNB3*. Conclusions: This pilot study demonstrates the feasibility of using FFPE samples from dogs for whole-genome sequencing and the detection of germline variants. Using FFPE samples in the analysis of suspected inherited diseases in domestic animals may represent a valuable approach in veterinary genetics if no other samples are available.

## 1. Introduction

Identifying variants in DNA sequences plays a fundamental role in uncovering the genetic basis of diseases and developing precision medicine approaches. Advances in sequencing technology have significantly reduced costs, making whole-genome sequencing (WGS) increasingly accessible and cost-effective. This has transformed WGS into a powerful tool for comprehensive genomic analysis, enabling the identification of a wide range of genetic variants, including single nucleotide variants (SNVs) and structural variants (SVs), across both research and clinical settings.

While DNA isolated from fresh frozen (FF) tissue remains the gold standard for WGS, archived formalin-fixed paraffin-embedded (FFPE) tissue samples are increasingly recognized as a valuable resource for genomic studies. Archived FFPE samples are widely available resources in human and veterinary medicine, preserving morphological characteristics while still enabling genomic or transcriptomic analyses. FFPE samples have been used to identify both somatic and germline SNVs across the genome [1]. Previous studies have shown a high concordance of analysis results between FFPE and FF tissue at sample [2] and variant levels [3,4].

Despite their value for genetic research, the formalin fixation and paraffin embedding processes in FFPE samples result in molecular damage to nucleic acids, which can interfere with genomic sequencing and variant analysis. For example, C:G>T:A substitutions arise from the dU:G and T:G mismatches induced by cytosine and 5-methylcytosine deamination, respectively. This and other molecular alterations due to fixation have been documented in various studies, revealing that FFPE samples tend to yield more genetic variants compared to their FF counterparts. Many of these variants are false positives, thereby complicating the interpretation of genomic data derived from FFPE tissues [5,6].

Unique molecular identifiers (UMIs) are short, random oligonucleotide tags added to each DNA fragment before any PCR amplification. As each starting DNA molecule gets a unique tag, all sequence reads sharing that tag are assumed to represent PCR duplicates that may be generated during library amplification steps. UMI-based sequencing has proven to be a highly effective method for increasing the precision in variant identification, with a particular strength in detecting low-frequency variants [7]. Combining FFPE sample processing with UMI-based sequencing enhances variant detection accuracy. UMIs enable grouping of reads and generation of consensus sequences for low-frequency variants. This approach filters out PCR-induced sequencing artifacts, improving the specificity of variant calls, especially for identifying low-frequency variants in heterogeneous samples [8].

Filtering FFPE-induced artificial variants from WGS data is crucial for accurate genomic analysis. Several methodologies in somatic variant calling have been developed to address this critical issue. Among these, deep learning models such as DEEPOMICS FFPE [9] have emerged, employing neural networks to differentiate between true variants and artifactual sequence changes. Additionally, combinatorial techniques that integrate multiple variant callers serve to mitigate the incidence of false-positive variants in FFPE samples [10]. Machine learning approaches, exemplified by FFPolish, utilize logistic regression to classify and filter out FFPE-specific false-positive variants [11]. Quality filters focusing on variant allele frequency, mapping quality, and allele depths further aid in distinguishing genuine variants from artifacts as recommended by GATK’s best practices protocol [12]. Moreover, strand bias detection techniques exploit the orientation differences in paired-end sequencing reads to identify FFPE-associated deaminations [13]. DeepVariant [14], a deep learning-based variant caller, uses convolutional neural networks to robustly distinguish true variants from FFPE artifacts, consistently providing high precision in both SNV and indel calling for challenging samples. These diverse methods aim to optimize the accuracy of variant calling in FFPE samples.

In this study, we analyzed three Weimaraner dog littermates affected by congenital mirror movement disorder 1 (CMM1). This inherited neurological disorder is caused by a frameshift variant in the *EFNB3* gene [15]. We compared illumina short-read sequencing data obtained from FF EDTA blood and FFPE tissue samples. We applied and compared two distinct approaches for variant calling: the Genome Analysis Toolkit (GATK) best practices pipeline [12] and DeepVariant, a deep learning neural network algorithm [14]. This comparative analysis highlights the strengths and limitations of traditional and contemporary computational techniques.

## 2. Materials and Methods

### 2.1. Genomic DNA Isolation and Library Preparation from FF Samples

Intact genomic DNA was extracted from frozen and thawed EDTA blood samples derived from three 2-month-old male dogs (dogs #1–3) using the Maxwell RSC Whole Blood Kit and the Maxwell RSC 48 instrument (Promega, Dübendorf, Switzerland). The size distribution of the DNA fragments was evaluated on a Femto Pulse capillary electrophoresis system (Agilent, Basel, Switzerland). Approximately 1 µg of genomic DNA was used to prepare PCR-free DNA libraries with the Illumina TruSeq PCR-free library preparation kit (Illumina, Zürich, Switzerland). The targeted insert size was 400–500 bp. The generation and analysis of the FF data was previously published [15].

### 2.2. Genomic DNA Isolation and Library Preparation from FFPE Samples

Genomic DNA from FFPE tissue samples of dogs #1–3 was isolated using the Maxwell RSC DNA FFPE Kit and the Maxwell RSC 48 instrument. We isolated DNA from kidney, skeletal muscle, and spleen samples. The size distribution of the DNA fragments was evaluated on a Femto Pulse capillary electrophoresis system. A total of 250 ng of spleen DNA was used to prepare sequencing libraries with the NEBNext UltraShear FFPE DNA Library Prep Kit (NEB #E6655; New England Biolabs, Ipswich, MA, USA).

### 2.3. Sequencing

Libraries were subjected to standard QC including concentration measurements by fluorimetry on a Qubit 4 fluorimeter (ThermoFisher, Basel, Switzerland) and fragment size analysis on a Fragment Analyzer 5200 (Agilent). Sequencing was performed on a NovaSeq 6000 instrument (Illumina). For FF libraries, ~200 million 2 × 150 bp read pairs were generated per dog. For FFPE libraries, ~450 million 2 × 100 bp genomic read pairs (R1/R3) were generated using a three-read configuration with a separate UMI-only read (R2), yielding three FASTQ files per sample (R1, R2-UMI, R3).

### 2.4. Variant Calling Pipelines

Preprocessing steps were identical for the GATK and DeepVariant pipelines (Figure 1). Raw FASTQ files were adapter-trimmed and quality-filtered with fastp (v0.23.2) [16]. For FF samples, Read 1 (R1) and Read 2 (R2) were used; for FFPE samples R1 and Read 3 (R3) were used for alignment. Reads were aligned to the dog reference genome UU_Cfam_GSD_1.0 using BWA-MEM2 (v2.2.0) [17], and coordinate-sorted BAMs were produced using samtools (v1.3) [18]. For FFPE data, unique molecular identifiers (UMIs) were added to the BAM using fgbio AnnotateBamWithUmis (v2.1.0) [19]. UMI-annotated BAMs from four sequencing lanes were merged with GATK MergeSamFiles [12]. Duplicate marking for FFPE was performed with Picard’s UmiAwareMarkDuplicatesWithMateCigar (GATK v4.4.0.0) [12]. For FF samples, after merging lanes, duplicates were marked with Picard (GATK v4.4.0.0) [12]. Base quality score recalibration (BQSR) was then applied using GATK (v4.4.0.0) [12].

Variant calling proceeded in parallel with GATK HaplotypeCaller and DeepVariant [14] (run_deepvariant), each producing gVCFs. Joint genotyping of GATK gVCFs used GenotypeGVCFs; DeepVariant gVCFs were jointly genotyped with GLnexus. GATK small-variant calls were hard-filtered as follows. For SNVs: QD < 2.0, QUAL < 30.0, SOR > 3.0, FS > 60.0, MQ < 40.0, MQRankSum < −12.5, ReadPosRankSum < −8.0. For indels: QD < 2.0, QUAL < 30.0, FS > 200.0, ReadPosRankSum < −20.0, SOR > 10. The DeepVariant vcf was not filtered, since the deep learning model used in DeepVariant is trained to identify true variants and differentiate them from sequencing errors, reducing the need for hard filtering. Finally, snpEff (v5.0e) [20] and NCBI Annotation Release 106 were used to annotate and functionally classify the variants, resulting in annotated VCF files.

### 2.5. Truth Set Construction

The GATK and DeepVariant shared SNV and indel calls from the FF data were used as a truth set to evaluate the accuracy of the variants called from matched FFPE tissues. Bcftools isec and merge functions were used to construct the truth set.

### 2.6. Variant Calling Evaluation

For each matched FF-FFPE sample pair, we performed sample-wise comparisons by extracting variants from both truth and query sets, filtering out homozygous reference calls (0/0), and using bcftools isec to identify intersections. The variant calls were evaluated using benchmarking metrics including true positives (variants found in both the FF truth set and FFPE calls), false negatives (variants in FF truth but missing in FFPE), false positives (variants in FFPE but not in FF truth), and genotype concordance.

### 2.7. Variant Filtering (Causal Variant for CMM1)

We had previously reported the identification of the causal variant for CMM1 from FF-derived WGS data [15]. To validate the usefulness of the FFPE-derived dataset, we used the vcf-file with the GATK FFPE variants in exactly the same way as in the original publication [15]. In brief, we extracted homozygous private variants for each of the three affected dogs. For these variants, an affected dog had to carry a homozygous alternate genotype (1/1), while the genomes from 1492 controls were required to either have a homozygous reference (0/0) or missing (./.) genotype. The controls contained samples from genetically diverse dog breeds, random-bred dogs, and wolves (Appendix A). However, the controls did not contain any other Weimaraner dogs to minimize the risk of accidentally including a carrier animal. Variants with a SnpEff-predicted effect of moderate or high were considered protein-changing variants as described [15].

## 3. Results

### 3.1. Quality Control of Genomic DNA from FF and FFPE Samples

This study was conducted with matched samples from three dogs that had been euthanized due to an inherited neurological disease. We previously reported the identification of the disease-causing variant in these dogs, *EFNB3*:XM_038536724.1:c.643_644dup. Causal variant identification had been achieved by the analysis of WGS data obtained from high-quality genomic DNA isolated from FF EDTA blood samples of the three dogs [15].

We used the opportunity of having access to FFPE tissue samples from several organs of the same three dogs to perform a methodological comparison between FF and FFPE samples. The FFPE samples had been routinely processed for pathological examinations and were stored for ~6 months at room temperature prior to DNA extraction. We isolated genomic DNA from the kidney, skeletal muscle and spleen. The yield and quality of the isolated DNA from the FFPE samples were comparable between the tested organs. DNA isolated from FFPE samples mostly consisted of fragments in the size range of 0–3000 bp with a maximum peak around 200 bp. In contrast, the genomic DNA from the FF samples contained much larger DNA fragments, with a majority of fragments in the size range between 2000 and 50,000 bp (Figure 2).

### 3.2. Library Preparation from FF and FFPE Samples

For FF DNA samples we generated PCR-free libraries with an average insert size of ~400 bp as PCR-free libraries can be expected to yield the most even coverage across the genome. Avoiding amplification steps during library preparation also minimized the risk of introducing artifactual sequence variants due to polymerase errors.

For the FFPE DNA samples, we chose the NEBNext UltraShear FFPE DNA Library Prep Kit, which employs an enzymatic fragmentation of the starting DNA. Enzymatic fragmentation methods are very effective but may lead to less even fragmentation than the mechanical shearing by ultrasound that was used with the FF samples. The reagents in the FFPE kit included a proprietary mix of enzymes optimized to repair damaged FFPE DNA. Due to the degradation of the starting input DNA, the average insert sizes were around ~200 bp in the FFPE libraries.

### 3.3. Sequencing FF and FFPE Samples

The FF libraries were sequenced with standard illumina 2 × 150 bp paired-end sequencing chemistry. Due to the smaller insert size of the FFPE libraries, we opted for 2 × 100 bp reads on these libraries. The targeted coverage was 20× for FF and 40× for FFPE libraries. The actually achieved sequencing metrics are shown in Table 1. The percentage of high-quality bases (Q20) was relatively consistent across all samples and types, indicating good overall sequencing quality.

### 3.4. Mapping FF and FFPE Samples

The filtered reads were mapped to the UU_Cfam_GSD_1.0 reference genome using bwa-mem2. Both FF and FFPE samples showed consistent alignment, with primary mapping rates exceeding 99% for all samples (Table 2). Low singleton rates (0.03–0.04%) confirmed good paired-end alignment across sample types. FFPE libraries displayed increased supplementary alignment (indicative of non-linear mappings) and a 2.1–3.6-fold higher number of inter-chromosomal read pairs, likely stemming from DNA fragmentation and crosslinking artifacts introduced by archival processing. Fresh samples displayed higher duplication rates (10.4–11.4%) using coordinate-based methods, whereas FFPE specimens achieved lower duplication levels (4.7–5.3%) through UMI-based deduplication, indicating a difference in library complexity. Soft-clip analysis further differentiated the groups: FF samples predominantly showed short clips (5–10 bp), while FFPE exhibited more variable distributions with longer clips (20–25 bp).

### 3.5. Variant Calling in FF Samples

WGS variant calling for FF samples was performed using GATK’s best practices pipeline as well as DeepVariant, which employs the run_deepvariant algorithm in place of GATK’s HaplotypeCaller (Figure 1). Table 3 summarizes the statistics of called variants from the two pipelines. Both pipelines identified 4,847,926 shared SNVs and 1,849,615 shared INDELs. Clear differences were seen among tool-specific calls: GATK identified more unique SNVs and INDELs than DeepVariant, along with a higher number of unique multiallelic sites for both variant types. The shared SNVs showed a transition-to-transversion (Ts/Tv) ratio of 2.09 which was close to the expected value for whole-genome sequencing. In contrast, variants unique to either GATK (Ts/Tv = 1.02) or DeepVariant (Ts/Tv = 1.03) showed lower ratios.

Likewise, the heterozygous-to-homozygous (het/hom) ratio of the shared set was 1.35. In contrast, GATK’s unique SNVs exhibited an extreme het/hom of 8.44 (almost all heterozygotes), and DeepVariant’s unique SNVs was at 0.55 (a strong bias toward homozygous alternates). Both variant calling methods reported similar frequencies of singleton variants.

For INDEL analysis, GATK again reports more unique INDELs than DeepVariant (342,648 vs. 157,599). The shared INDELs display a balanced deletion-to-insertion ratio of about 1.04, as expected in genome-wide variant calling. In contrast, GATK’s private INDELs have a strong deletion bias (del/ins ≈ 1.22), whereas DeepVariant’s private set is insertion-skewed (del/ins ≈ 0.58). Both tools identified 180,000 shared multiallelic INDEL sites, with GATK contributing 81,000 additional unique multiallelic calls and DeepVariant contributing 62,000.

### 3.6. Performance Evaluation of DeepVariant Against GATK Benchmark in FF Samples

Benchmarking DeepVariant SNV calls against GATK revealed a high degree of genotype concordance, with values consistently above 98% for all three dog samples (Table 4). For SNVs, genotype concordance exceeded 98.8% in all cases, with F1 scores ranging from 95.98% to 96.20%, indicating strong agreement between the Deepvariant calls and the GATK truth set. Precision values above 96% reflected a low false-positive rate, while recall values consistently above 95% showed that most GATK variants were successfully captured. True positives were high for all dogs, and ranged from 3,979,168 to 3,985,449. False-positive rates were 3.6–3.7% and false-negative rates were 4.0–4.3%.

However, the performance metrics for insertions and deletions (INDELs) showed lower values compared to SNVs, with average concordance of 84.89% and average F1 scores of 83.79%. While indel precision remained relatively high (87.09% average), recall dropped significantly (80.78% average).

### 3.7. FFPE WGS Variant Calls

We compared GATK and DeepVariant pipelines using WGS data from FFPE samples (Table 5). Both callers jointly identified 4.5 million concordant SNVs with a Ts/Tv ratio of 2.11. GATK uniquely called 1,976,503 SNVs and 950,403 INDELs, while DeepVariant uniquely called 102,914 SNVs and 267,573 INDELs. Uniquely called variants displayed reduced Ts/Tv ratios (1.25 and 0.87) relative to shared calls, with GATK showing more transitions in comparison to DeepVariant. Unique GATK SNVs displayed an unusually high het/hom ratio of 20.50. For INDELs, deletion-to-insertion ratios were 1.14 for shared variants, 1.05 for GATK-unique variants, and 0.76 for DeepVariant-unique variants. GATK identified more multiallelic sites than DeepVariant for both SNVs (51,720 vs. 262) and INDELs (511,355 vs. 163,195) and also significantly more singleton variants.

### 3.8. High-Confidence Truth Set from FF Whole-Genome Sequencing

A truth set was constructed from variants called by both GATK and DeepVariant in FF samples. This intersection yielded 4,847,926 shared SNPs and 1,849,615 shared INDELs (Table 3). The shared variants were used as a putative truth set, based on the rationale that concordance between independent variant callers minimized caller-specific artifacts. Moreover, key quality metrics, including the Ts/Tv ratio and the del/ins ratio, were consistent with expected genomic values, further supporting the reliability of this set.

### 3.9. Evaluation of FFPE WGS Variant Calls Using Truth Set

GATK achieved higher SNV recovery rates (93–98%) compared to DeepVariant (84–92%) across all samples (Table 6). However, DeepVariant demonstrated substantially lower false-positive rates for SNVs (3.1–3.2%) versus GATK (14–21%). For INDELs, both platforms showed similar recovery performance (GATK: 81–85%, DeepVariant: 73–81%), but DeepVariant again exhibited lower false-positive rates (13–14%) compared to GATK (27–30%). Dog #2 consistently showed the poorest performance across both variant types and platforms, with the lowest recovery rates and highest false-negative rates. DeepVariant maintained more consistent false-positive rates across samples, while GATK’s false-positive rates varied considerably between dogs.

### 3.10. Genotype Concordance

Comparison of genotype concordance between FFPE-derived calls and the FF truth sets again showed consistently higher performance for SNVs than for INDELs across all three samples (Table 7). Using GATK, SNV concordance ranged from 93.6% to 96.0%, while DeepVariant achieved lower concordance, between 79.5% and 80.7%.

For INDELs, however, the opposite trend was observed. Concordance with GATK was relatively low, ranging between 35.1% and 35.6%, while DeepVariant performed better in this category, achieving 43.0–43.7% concordance across samples.

### 3.11. Characteristics of FFPE-Induced Artifacts

We compared single-nucleotide substitution spectra between matched fresh frozen (FF) and FFPE samples using the two callers. GATK detected higher variant rates in FFPE samples across all 12 substitution classes, with transition types (C>T, G>A, A>G, T>C) showing the largest increases (15–20% higher than FF). DeepVariant showed the reverse pattern, with FFPE samples exhibiting lower call rates than FF across all classes, particularly for transitions (8–15% reduction; Figure 3).

### 3.12. Identification of the Causal Variant for CMM1 from the FFPE Data

In order to test the usefulness of the FFPE dataset, we attempted to identify the causal variant for CMM1 using the same filtering steps as we had previously reported with data from FF samples [15]. The FFPE dataset yielded many more private variants than the FF dataset, including four shared protein-changing private variants on autosomes (Table 8). A visual inspection of the short-read alignments at the remaining four variants revealed that three of them were due to technical artifacts (incorrect variant calls in the FFPE data) and that only the *EFNB3* variant represented a true variant. The false-positive variants also did not localize to the critical interval. Thus, combining the variant filtering with the positional information from linkage and autozygosity mapping would also have allowed the straightforward identification of the correct causal variant. More details on the quality of the data and some examples of concordantly and discordantly called variants are shown in Appendix A.

## 4. Discussion

We compared variant calls derived from FF EDTA blood samples with matched FFPE spleen tissue samples from three Weimaraner dogs. We used two current standard algorithms for calling the variants for the comparison. GATK is a widely used, robust toolkit for variant calling known for its accuracy and continuous updates, making it a gold standard for analyzing next-generation sequencing data. DeepVariant, on the other hand, leverages a deep learning approach to identify genetic variations from aligned reads, often demonstrating superior performance in complex genomic regions.

Matched FF and FFPE WGS from three Weimaraner littermates showed the expected fixation effects: FFPE DNA was highly fragmented (peak ~200 bp), and FFPE alignments displayed more supplementary mappings and longer soft-clips, while UMI-based deduplication reduced duplicates relative to FF. The observed differences in sequencing metrics between FF and FFPE samples align with the existing literature on the topic [21,22].

GATK demonstrated superior sensitivity for both SNVs and INDELs when calling variants from FFPE data, recovering a higher fraction of presumed true FF variants than DeepVariant. However, this increased recall was accompanied by a substantial trade-off in specificity, as false-positive rates were markedly higher for GATK, especially for SNVs (up to 21%, versus ~3% for DeepVariant). DeepVariant, in contrast, maintained lower false-positive rates but at the cost of missing more true positives, especially for challenging INDELs. Notably, one FFPE sample (dog #2) consistently underperformed across all tools and variant classes, highlighting the impact of sample quality or individual-specific factors on variant detection accuracy.

The genotype concordance results reveal that the two algorithms exhibited opposite strengths depending on variant type, reflecting fundamental differences in their calling strategies. GATK’s superior SNV concordance (93.6–96.0% versus DeepVariant’s 79.5–80.7%) aligns with its more permissive approach that prioritizes sensitivity, successfully capturing more true variants despite generating additional false positives. Conversely, DeepVariant’s higher INDEL concordance (43.0–43.7% versus GATK’s 35.1–35.6%) suggests that its machine learning framework is better equipped to handle the complex alignment patterns typical of insertion–deletion events, where traditional statistical models often struggle.

The substitution spectra analysis also provides mechanistic insight into these performance differences: GATK’s inflation of all variant classes, particularly C>T and G>A transitions characteristic of FFPE-induced cytosine deamination, indicates that the algorithm may be overcalling artifacts as true variants. In contrast, DeepVariant’s global suppression of calls, with strongest reduction in transitions, demonstrates that its deep learning model has been trained to recognize and down-weight deamination-like signals that are hallmarks of FFPE processing artifacts.

Despite elevated artifact burden in FFPE samples, the analytical pipeline successfully identified the true causal *EFNB3* variant when combined with careful subjective expert review. Integration of positional mapping data could further enhance variant prioritization. These findings demonstrate that stringent filtering combined with contextual genomic annotation enables robust detection of clinically relevant variants even under high false discovery rates typical of FFPE sequencing.

This study shows encouraging results, but not without drawbacks. The study’s generalizability is constrained by the small cohort size. Moreover, reliance on an FF-derived truth set and limited external validation means some performance estimates may reflect truth set limitations rather than algorithmic differences. Recent comparative studies have shown that benchmarking across variant calling algorithms frequently reveals substantial discordance, emphasizing that relying on a single tool may risk missing true variants or overcalling artifacts, particularly in FFPE-derived datasets [6]. While emerging machine learning and combinatorial methods are improving accuracy, a multi-algorithm approach or expert curation remains essential for clinically reliable outcomes [11].

Despite all the limitations and challenges of working with FFPE samples, the successful identification of the published *EFNB3* variant [15] from FFPE WGS data demonstrates the reliability of variant calling using whole-genome sequencing from FFPE material.

## 5. Conclusions

This pilot study compared WGS data from matched FF and FFPE samples of three dogs. Using currently available software for analysis, it was possible to correctly identify a specific germline variant from the FFPE data causing a monogenic inherited disease. However, while canine FFPE samples can be successfully used, their analysis poses a number of challenges and data quality from FFPE samples is clearly inferior to data from FF samples. This study provides a transparent documentation of an FFPE WGS analysis in veterinary genetics and will provide a benchmark for future related studies.

## Figures and Tables

**Figure 1 genes-16-01371-f001:**
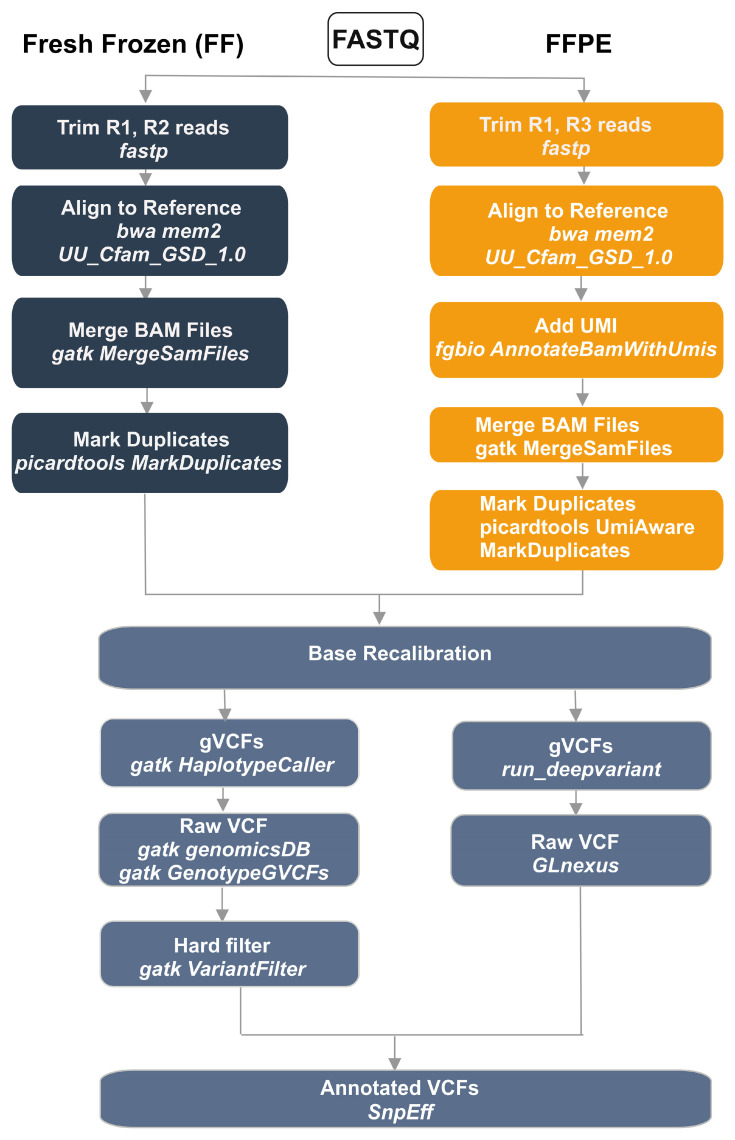
Bioinformatics pipeline for genomic variant calling using GATK and DeepVariant in fresh frozen (FF) and formalin-fixed paraffin-embedded (FFPE) samples. FF samples (dark blue) undergo standard R1, R2 read trimming, while FFPE samples (orange) require R1, R3 trimming (see methods). Both workflows merge at base quality score recalibration (blue-gray indicates common processing steps). The pipeline then splits into parallel variant calling using GATK HaplotypeCaller or DeepVariant, generating gVCFs that are processed through genomicsDB/GenotypeGVCFs and GLnexus, respectively. Raw VCFs undergo hard filtering via GATK VariantFilter before final annotation with snpEff. Tool names are indicated in italics below each processing step.

**Figure 2 genes-16-01371-f002:**
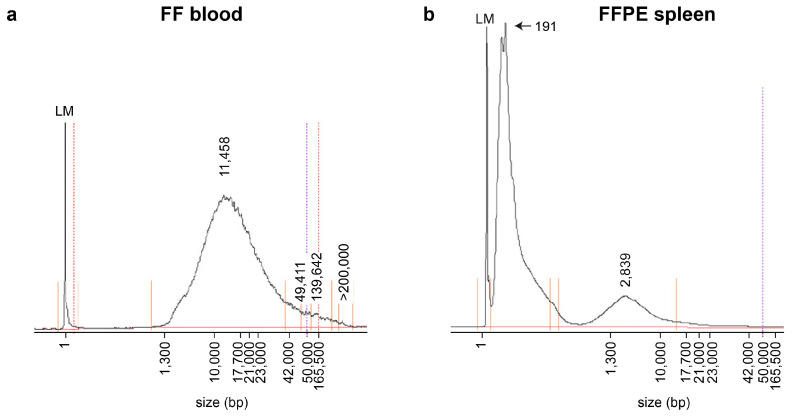
Fragment size analysis of input genomic DNA for library construction on an automated capillary gel electrophoresis instrument (Femto Pulse system). (**a**) Genomic DNA isolated from an FF EDTA blood sample of dog #1. (**b**) Genomic DNA isolated from an FFPE spleen sample of dog #1. Note the much smaller fragment sizes in the DNA sample isolated from FFPE tissue.

**Figure 3 genes-16-01371-f003:**
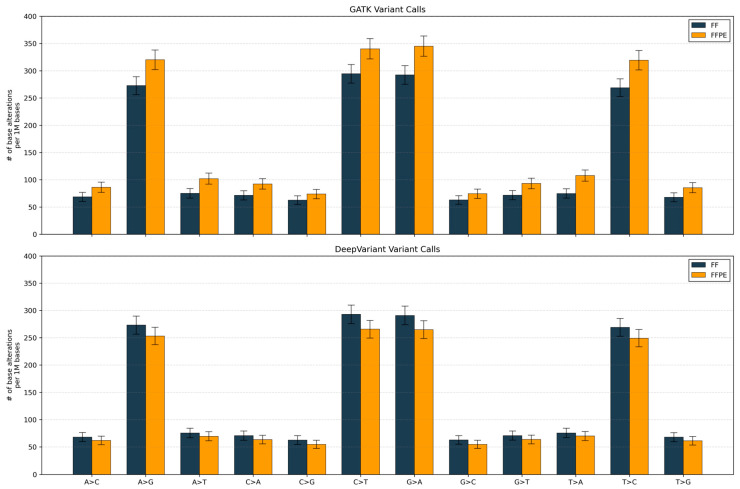
Comparison of single nucleotide substitution rates in FF vs. FFPE samples. Bars show mean calls per 10 Mb stratified by substitution class for GATK (**top**) and DeepVariant (**bottom**); FF (dark blue) and FFPE (orange). Error bars denote Poisson standard errors (√count).

**Table 1 genes-16-01371-t001:** Sequencing statistics for FF and FFPE libraries.

Metric	Type	Dog #1	Dog #2	Dog #3
Total reads	FF	412,369,436	480,688,536	403,306,286
	FFPE	941,485,714	952,830,830	950,777,134
Total reads (after filtering)	FF	404,941,234	469,900,742	397,195,242
	FFPE	934,167,479	942,411,639	945,296,080
Q20 bases (after filtering)	FF	97.48%	96.99%	97.23%
	FFPE	97.34%	97.36%	97.49%
Coverage	FF	25.35×	29.42×	24.87×
	FFPE	38.35×	37.88×	38.83×

**Table 2 genes-16-01371-t002:** Mapping and alignment metrics for FF and FFPE libraries.

Metric	Type	Dog #1	Dog #2	Dog #3
Primary mapping	FF	403,234,828 (99.57%)	467,919,136 (99.57%)	395,542,014 (99.58%)
	FFPE	929,045,634 (99.45%)	934,845,958 (99.20%)	939,167,618 (99.35%)
Supplementary mapping rate	FF	0.42%	0.42%	0.42%
	FFPE	0.55%	0.80%	0.65%
Duplicate rate	FF	11.37%	10.40%	11.28%
	FFPE	4.71%	4.84%	5.29%
Properly paired	FF	98.35%	98.04%	98.63%
	FFPE	97.76%	97.99%	98.08%
Different Chr mapping	FF	0.98%	1.25%	0.71%
	FFPE	1.05%	0.55%	1.05%

**Table 3 genes-16-01371-t003:** Comparative analysis of FF variant calling metrics: GATK vs. DeepVariant.

Metric	Unique to GATK	Unique to DeepVariant	Shared
**SNVs**			
Total SNVs	235,281	195,446	4,847,926
Multiallelic SNVs	31,524	349	3061
Ts/Tv ratio	1.00	1.01	2.09
Singleton SNVs	34,696	27,624	629,490
het/hom ratio *	8.44	0.55	1.35
**INDELs**			
Total INDELs	342,648	157,599	1,849,615
Multiallelic sites	81,013	61,650	180,348
Deletions (del)	184,152	56,262	937,206
Insertions (ins)	151,227	97,685	898,956
del/ins ratio	1.22	0.58	1.04
Singleton INDELs	50,715	36,929	227,304

* Averaged het/hom ratio across the three samples.

**Table 4 genes-16-01371-t004:** SNV and INDEL calling metrics in FF samples.

Metric	Type	Dog #1	Dog #2	Dog #3
Total SNVs	Truth	4,150,703	4,158,502	4,158,848
	Query	4,135,339	4,132,691	4,132,647
Genotype Concordance		98.88%	98.82%	98.86%
True Positives		3,985,449	3,982,023	3,979,168
False Positives		149,890	150,668	153,479
False Negatives		165,254	176,479	179,680
Precision		96.38%	96.35%	96.29%
Recall		96.02%	95.76%	95.68%
F1 Score		96.20%	96.05%	95.98%
Total INDELs	Truth	1,909,453	1,915,452	1,903,426
	Query	1,782,062	1,784,162	1,773,434
Genotype Concordance		85.24%	85.61%	85.25%
True Positives		1,563,156	1,577,855	1,566,681
False Positives		218,906	206,307	206,753
False Negatives		346,297	337,597	336,745
Precision		87.72%	88.44%	88.34%
Recall		81.86%	82.38%	82.31%
F1 Score		84.69%	85.30%	85.22%

**Table 5 genes-16-01371-t005:** Comparative analysis of FFPE variant calling metrics: GATK vs. DeepVariant.

Metric	Unique to GATK	Unique to DeepVariant	Shared
**SNVs**			
Total SNVs	1,976,503	102,914	4,495,239
Multiallelic sites	51,720	262	2461
Transitions (Ts)	1,089,065	47,827	3,050,681
Transversions (Tv)	856,685	55,087	1,444,558
Ts/Tv ratio	1.25	0.87	2.11
Singleton SNVs	1,363,683	29,399	640,424
het/hom ratio *	20.50	0.56	0.96
**INDELs**			
Total INDELs	950,403	267,573	1,516,200
Multiallelic sites	511,355	163,195	727,055
Deletions (del)	475,018	113,243	801,182
Insertions (ins)	454,132	149,605	705,462
del/ins ratio	1.05	0.76	1.14
Singleton INDELs	550,666	64,493	223,781

* Averaged het/hom ratio across the three samples.

**Table 6 genes-16-01371-t006:** Evaluation metrics for FFPE variant calls against a truth set from FF tissue.

Metric	Dog #1	Dog #2	Dog #3
**GATK Evaluation**			
**SNVs**			
Truth set	4,000,130	4,002,055	3,995,612
Recovered in FFPE	3,921,247 (98.02%)	3,727,751(93.14%)	3,811,972 (96.44%)
False negatives in FFPE	78,883 (1.97%)	274,304 (6.85%)	142,183 (3.55%)
False positives in FFPE	648,346 (14.19%)	995,747 (21.08%)	807,903 (17.33%)
**INDELs**			
Truth set	1,560,501	1,562,339	1,554,240
Recovered in FFPE	1,329,857 (85.21%)	1,268,732 (81.20%)	1,293,729 (83.23%)
False negatives in FFPE	230,644 (14.78%)	293,607 (18.79%)	260,511 (16.76%)
False positives in FFPE	490,105 (26.93%)	545,605 (30.07%)	506,477 (28.14%)
**DeepVariant Evaluation**			
**SNVs**			
Truth set	4,000,130	4,002,055	3,995,612
Recovered in FFPE	3,666,386 (91.65%)	3,350,023 (83.70%)	3,541,117 (88.62%)
False negatives in FFPE	333,744 (8.34%)	652,032 (16.29%)	454,495 (11.37%)
False positives in FFPE	118,767 (3.14%)	108,750 (3.14%)	116,733 (3.19%)
**INDELs**			
Truth set	1,560,501	1,562,339	1,554,240
Recovered in FFPE	1,259,519 (80.71%)	1,147,621 (73.45%)	1,211,533 (77.95%)
False negatives in FFPE	300,982 (19.28%)	414,718 (26.54%)	342,707 (22.04%)
False positives in FFPE	208,727 (14.22%)	182,160 (13.70%)	198,377 (14.07%)

**Table 7 genes-16-01371-t007:** Genotype concordance for truth sets from FF tissue.

Metric	Dog #1	Dog #2	Dog #3
**SNVs**			
Concordant genotypes (GATK)	3,764,256 (96%)	3,487,881 (93.6%)	3,677,866 (95.4%)
Concordant genotypes (DeepVariant)	2,961,894 (80.7%)	2,664,263 (79.5%)	2,847,705 (80.4%)
**INDELs**			
Concordant genotypes (GATK)	473,905 (35.6%)	452,145 (35.6%)	455,234 (35.1%)
Concordant genotypes (DeepVariant)	551,933 (43.7%)	495,256 (43.1%)	521,854 (43.0%)

**Table 8 genes-16-01371-t008:** Results of variant filtering in three affected dogs against 1489 control genomes. Only homozygous variants are reported. The data from the FF dataset were previously reported in [15].

FF Dataset	Dog #1	Dog #2	Dog #3	Shared
Total variants in whole genome	2,911,727	2,929,067	2,960,172	1,938,789
Private variants in whole genome	839	737	902	217
Private protein-changing variants in whole genome	2	3	3	1
Private protein-changing variants in critical interval ^1^	1	1	1	1
**FFPE Dataset (GATK)**				
Total variants in whole genome	2,560,769	2,583,318	2,595,726	1,599,000
Private variants in whole genome	66,613	101,905	60,483	44,911
Private protein-changing variants in whole genome	124	187	130	87
Private protein-changing variants on autosomes ^2^	17	68	19	4
Private protein-changing variants in critical interval ^1^	1	1	1	1

^1^ Linkage and autozygosity mapping in the original family had delineated a critical interval of 43.2 Mb comprising 5 segments on 5 different chromosomes [15]. ^2^ Variants on the sex chromosomes and unplaced contigs were excluded.

## Data Availability

The FF whole-genome sequence data can be found under the study accession PRJEB16012 and sample accessions SAMEA114382052 (dog #1), SAMEA114382053 (dog #2), and SAMEA114382054 (dog #3) at the European Nucleotide Archive. The FFPE whole-genome sequence data can be found under the study accession PRJEB85286 and sample accessions SAMEA117639516 (dog #1), SAMEA117639517 (dog #2), and SAMEA117639518 (dog #3). All accessions including the 1489 control genomes are given in Appendix A.

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
