# Peer review of "Germline Variant Call Accuracy in Whole Genome Sequence Data from Canine Formalin-Fixed Paraffin-Embedded Tissue Samples"

_genes, 2025, doi:10.3390/genes16111371_

Round 1

Reviewer 1 Report

Comments and Suggestions for Authors

The review of the mansucript titles: Germline Variant Call Accuracy in Whole Genome Sequence Data from Canine FFPE Tissue Samples submitted to Genes

The study presents a valuable evaluation of the feasibility of using formalin-fixed paraffin-embedded (FFPE) samples for whole genome sequencing (WGS) in dogs, with a specific focus on the detection of germline variants. The authors effectively demonstrate that, despite the expected DNA degradation and technical challenges associated with FFPE material, it is still possible to obtain sequencing data of sufficient quality to identify disease-causing mutations. The successful detection of the EFNB3 variant from FFPE-derived data represents a convincing proof of concept. The authors are aware of the limitations of this work - very limited sample size and the lack of external validation. The authors rightly note that the reliance on a small cohort limits generalizability, and further studies with larger sample sets would be necessary to confirm the robustness of the approach. Nevertheless, as a pilot investigation, the work offers a meaningful contribution to veterinary genomics by demonstrating that archived FFPE samples can serve as a viable alternative when fresh frozen material is unavailable. The comparison of two variant-calling tools (GATK and DeepVariant) is presented in a clear way. The technical aspects are described in sufficient detail. Overall, this is a well-executed and informative study. Although modest in scale, it addresses an important methodological question and offers practical insights for researchers working with limited or degraded genetic material in animal genetics.

My only minor comments concern the mistakes in Table numbering:

Line 292 – this should be Table 6? – this table is not indicated in the text

Lines 297 and 302 – this should be Table 7?

Lines 319 and 327 – this should be Table 8?

Author Response

0.

The study presents a valuable evaluation of the feasibility of using formalin-fixed paraffin-embedded (FFPE) samples for whole genome sequencing (WGS) in dogs, with a specific focus on the detection of germline variants. The authors effectively demonstrate that, despite the expected DNA degradation and technical challenges associated with FFPE material, it is still possible to obtain sequencing data of sufficient quality to identify disease-causing mutations. The successful detection of the EFNB3 variant from FFPE-derived data represents a convincing proof of concept. The authors are aware of the limitations of this work - very limited sample size and the lack of external validation. The authors rightly note that the reliance on a small cohort limits generalizability, and further studies with larger sample sets would be necessary to confirm the robustness of the approach. Nevertheless, as a pilot investigation, the work offers a meaningful contribution to veterinary genomics by demonstrating that archived FFPE samples can serve as a viable alternative when fresh frozen material is unavailable. The comparison of two variant-calling tools (GATK and DeepVariant) is presented in a clear way. The technical aspects are described in sufficient detail. Overall, this is a well-executed and informative study. Although modest in scale, it addresses an important methodological question and offers practical insights for researchers working with limited or degraded genetic material in animal genetics.

Response: We thank the reviewer for the overall positive assessment.

1.

Line 292 – this should be Table 6? – this table is not indicated in the text

Response: Thank you for spotting the error. We revised the numbering of the tables accordingly.

2.

Lines 297 and 302 – this should be Table 7?

Response: Revised accordingly.

3.

Lines 319 and 327 – this should be Table 8?

Response: Revised accordingly.

Reviewer 2 Report

Comments and Suggestions for Authors

Summary:

In this article, the authors compared whole-genome sequencing data from Fresh frozen (FF) and formalin-fixed paraffin embedded (FFPE) specimens of three Weimaraner dogs. They have used several bioinformatics software such as Genome Analysis Toolkit and DeepVariant for their research. Therefore, they were capable of identifying a specific germline variant from the FFPE data responsible a monogenic inherited disease. The research is pertinent, and theory based. However, there are some comments that need to be addressed.

Major comments

  1. In relation to the title of the manuscript, it is recommended that 'FFPE' is changed to 'formalin-fixed paraffin-embedded', since the title of the article should be precise and avoid abbreviations to ensure better understanding and accessibility for the reader.

  2. The introduction section should be extended to include more bibliography related to congenital mirror movement disorder 1 (EFNB3-related) and the software used in this research.

  3. Several abbreviations in the manuscript are not defined. It is recommended that they are all defined.

  4. The authors used several softwares for their research. It is highly recommended that links to this software are added to the manuscript.

  5. In the Materials and Methods section, it is highly recommended that the gender and age of the dogs studied are included.

  6. The authors should discuss more their results in relation to other recent scientific research.

Author Response

0.

In this article, the authors compared whole-genome sequencing data from Fresh frozen (FF) and formalin-fixed paraffin embedded (FFPE) specimens of three Weimaraner dogs. They have used several bioinformatics software such as Genome Analysis Toolkit and DeepVariant for their research. Therefore, they were capable of identifying a specific germline variant from the FFPE data responsible a monogenic inherited disease. The research is pertinent, and theory based. However, there are some comments that need to be addressed.

Response: We thank the reviewer for the overall positive assessment.

1.

In relation to the title of the manuscript, it is recommended that 'FFPE' is changed to 'formalin-fixed paraffin-embedded', since the title of the article should be precise and avoid abbreviations to ensure better understanding and accessibility for the reader.

Response: Revised accordingly.

  1.  

The introduction section should be extended to include more bibliography related to congenital mirror movement disorder 1 (EFNB3-related) and the software used in this research.

Response: Congenital mirror movement disorder 1 (CMM1) was only recently discovered and characterized. There is only one published paper on this disease, which we already cited (new ref. 15, Schwarz et al. 2025).

We added more details and references on the software to the introduction (lines 73-78).

3.

Several abbreviations in the manuscript are not defined. It is recommended that they are all defined.

Response: We had hoped that all unusual abbreviations are sufficiently explained. Unfortunately, the reviewer did not point out specific instances, for which an explanation is missing. We are happy to add more explanations, if specific abbreviations are indicated during the production stage (correction of page proofs).

4.

The authors used several softwares for their research. It is highly recommended that links to this software are added to the manuscript.

Response: We added additional references for the software (new references 16-20).

5.

In the Materials and Methods section, it is highly recommended that the gender and age of the dogs studied are included.

Response: Revised accordingly.

6.

The authors should discuss more their results in relation to other recent scientific research.

Response: We added more text and an additional reference to the discussion (lines 382-387).

Round 2

Reviewer 2 Report

Comments and Suggestions for Authors

The modifications introduced by the authors are satisfactory.